# The Neural Testbed:
# Evaluating Joint Predictions

**Ian Osband**,* **Zheng Wen, Seyed Mohammad Asghari,**
**Brendan O'Donoghue, Botao Hao, Dieterich Lawson, Morteza Ibrahimi,**
**Xiuyuan Lu, Vikranth Dwaracherla, and Benjamin Van Roy**
DeepMind, Efficient Agent Team, Mountain View

## Abstract

Predictive distributions quantify uncertainties ignored by point estimates. This paper introduces *The Neural Testbed*: an open-source benchmark for controlled and principled evaluation of agents that generate such predictions. Crucially, the testbed assesses agents not only on the quality of their marginal predictions per input, but also on their joint predictions across many inputs. We evaluate a range of agents using a simple neural network data generating process. Our results indicate that some popular Bayesian deep learning agents do not fare well with joint predictions, even when they can produce accurate marginal predictions. We also show that the quality of joint predictions drives performance in downstream decision tasks. We find these results are robust across choice a wide range of generative models, and highlight the practical importance of joint predictions to the community.

## 1 Introduction

Most work on supervised learning has focused on marginal predictions. Marginal predictions predict one label given one input, but do not model the dependence between multiple predictions. For decision making, it is not enough to have good marginal predictions; the quality of *joint* predictions drives decision performance (Wen et al., 2022). Joint predictions predict multiple labels given multiple inputs, and may capture some correlation between outcomes. This distinction can be particularly important in learning settings where joint predictions allow an agent to distinguish what it knows from what it does not know (Li et al., 2011; Lu et al., 2021).

Figure 1: Two coins with identical marginal predictions, but distinguished by joint predictions.

---

*Contact `iosband@deepmind.com`

36th Conference on Neural Information Processing Systems (NeurIPS 2022).

Figure 1 presents a stylized example designed to highlight the importance of joint predictions in decision making. Consider two coins '£' and '$' with different *bias*='probability of heads'. Coin £ has a known bias of $\frac{1}{2}$, whereas coin $ has an unknown bias of either 0 or 1, and which are both equally likely. Examining the marginal prediction over a single flip: the two coins present identical outcomes 50:50. However, if we consider the outcome over two successive flips, which can be modeled as a two-by-two grid, then the difference between these coins is evident in their joint predictions. If you want to maximize the cumulative heads through time, then it's important to know the difference between these two settings. In this case, a learning agent should first choose $ and then, depending on the outcome of that flip heads/tails, employ a fixed policy of $/£ forward. Marginal predictions alone cannot drive this sort of policy, since they do not distinguish the two coins (Wen et al., 2022).

Our research is motivated by the grand challenges in artificial intelligence, and the great progress that has been made in deep learning systems (Krizhevsky et al., 2012; Brown et al., 2020). However, as these systems move beyond prediction and towards actually making decisions we have very little understanding of how and where popular deep learning approaches are suitable for joint predictions and hence decision making (Mnih et al., 2015; Silver et al., 2016). To this end, we introduce *The Neural Testbed* as a simple and clear benchmark for evaluating the quality of joint predictions in deep learning systems. This work is meant to be a 'sanity check' for popular deep learning approaches in a simple setting, and one that can help guide future research.

The Neural Testbed works by generating random classification problems using a neural-network-based generative process. The testbed splits data into a training set and testing set, allows a deep learning agent to train on the training set, and then evaluates the quality of the predictions on the testing set. It is worth noting that the problem framed by the Testbed is a *computational* one. Optimal performance would be attained by carrying out exact Bayesian inference: given infinite compute time, an agent could calculate the posterior distribution, which maximizes performance. However, due to the complexity of the data generating process, this is infeasible. The agents we study serve as approximate inference algorithms, and we can compare their performance purely through the quality of their predictions, without worrying 'is XYZ Bayesian?' (Izmailov et al., 2021).

Figure 2 offers a preview of our results in Section 4, where we compare benchmark approaches to Bayesian deep learning. This plot shows the KL loss when making $\tau$ simultaneous predictions. We compare the quality of marginal ($\tau = 1$) and joint ($\tau = 10$) predictions, normalized so that and MLP has loss=1. We see that, after tuning, most Bayesian deep learning approaches do not significantly outperform a single MLP in marginal predictions.

However, once we examine joint predictive distributions of order $\tau = 10$, there is a clear difference in performance among benchmark agents. In particular, some of the most popular benchmark approaches to Bayesian deep learning (`ensemble` (Lakshminarayanan et al., 2017), `dropout` (Gal and Ghahramani, 2016), `bbb` (Blundell et al., 2015)) do not outperform the baseline MLP when evaluated in joint predictions. At the same time, there are other approaches that perform much better in terms of joint predictions in this simple synthetic challenge. We will go on to show that these same agents perform better in decision making, and that these observations are robust to choice of generative model.

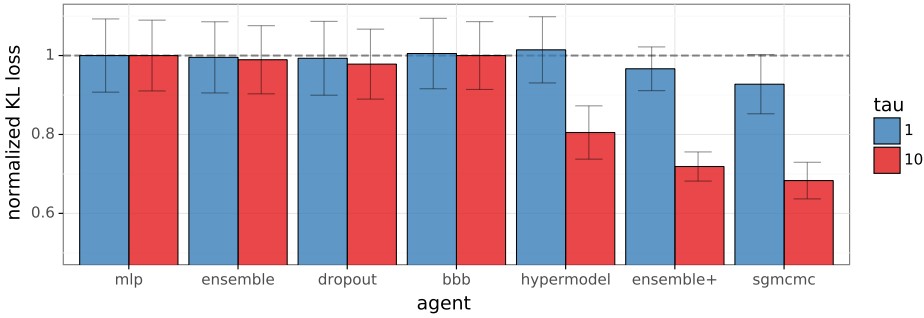

Figure 2: Quality of marginal and joint predictions on Neural Testbed (Section 4.2).

## 1.1 Key contributions

**We introduce *The Neural Testbed*, a simple benchmark for the field that involves making predictions in a neural-network-based generative model.** This work helps to bridge theory and practice, and provide an objective metric to assess the quality of approximate posterior inference in neural networks. We are the first paper to propose a concrete evaluation procedure for the quality of joint predictions in neural network classification.

**Together with this conceptual contribution, we open-source code in Appendix A.** This consists of highly optimized evaluation code, reference agent implementations and automated reproducible analysis. The testbed uses JAX internally (Bradbury et al., 2018), but can be used to evaluate any python agent. We believe that this library will be a major contribution to researchers and, due to its low computational cost, a boon to accessibility.

We use this new benchmark to obtain some important new experimental results. **We discover that several of the most popular approaches to Bayesian deep learning do not perform well at joint prediction**, and highlight this issue to the community. Further, we show that there *are* alternative approaches that *do* perform well in terms of joint prediction. Prior work has suggested that, in theory, the quality of joint predictions can drive decision performance (Wen et al., 2022). In this paper we provide empirical evidence that this effect occurs in practical deep learning systems. **We observe that performance in a neural bandit is highly correlated with performance in joint prediction**, and that it is not significantly correlated with the quality of marginal predictions.

Finally, **we show that the results in this paper are robust to the variations in the data generating model**. Although we focus on a 2-layer ReLU MLP with 50 hidden units for most of our experiments, the results we obtain are highly correlated across a wide range of alternative activation functions or network widths. This robustness supports the view that the field should be aware of these issues in joint prediction, and may help to stimulate future research in this area. Follow-up work has gone on to show that these results also carry over to challenge datasets popular in the community (Osband et al., 2022).

## 1.2 Related work

There is a rich literature around uncertainty estimation in deep learning. Much of this work has focused on agent development, with a wide variety of approaches including variational inference (Blundell et al., 2015), dropout (Gal and Ghahramani, 2016), ensembles (Osband and Van Roy, 2015; Lakshminarayanan et al., 2017), and MCMC (Welling and Teh, 2011; Hoffman et al., 2014). However, even when approaches become popular within particular research communities, there are still significant disagreements over the quality of the resultant uncertainty estimates (Osband, 2016; Hron et al., 2017).

Bayesian deep learning has largely relied on benchmark problems to guide agent development and measure agent progress. These typically include classic deep learning datasets but supplement the usual goal of classification accuracy to include an evaluation of the probablistic predictions via negative log likelihood (NLL) and expected calibration error (ECE) (Nado et al., 2021). More recently, several efforts have been made to supplement these datasets with challenges tailored towards Bayesian deep learning, and explicit Bayesian inference (Wilson et al., 2021). This literature has largely focused on evaluating marginal predictions, paired with evaluation on downstream tasks (Riquelme et al., 2018). Our work is motivated by the importance of *joint* predictions in driving good performance in sequential decisions (Wen et al., 2022). We share motivation with the work of Wang et al. (2021), but show that directly measuring joint likelihoods can provide new information beyond marginals. Follow up work has built upon the research in our paper, to extend the analysis of joint distributions to higher-order joint distributions, and empirical datasets (Osband et al., 2022).

## 2 Evaluating predictive distributions

In this section, we introduce notation for the standard supervised learning we consider as well as our evaluation metric: KL-loss. We review the distinction between marginal and joint predictions, and numerical schemes to estimate KL divergence via Monte Carlo sampling.

**Algorithm 1** KL-Loss Estimation

**for** $j = 1, 2, \ldots, J$ **do**
    sample environment and training data
    train agent on training data
    **for** $n = 1, 2, \ldots, N$ **do**
        sample $\tau$ test data pairs
        compute environmentlikelihood $p_{j,n}$
        compute agent likelihood $\hat{p}_{j,n}$
    **end for**
**end for**
**return** $\frac{1}{JN} \sum_{j=1}^{J} \sum_{n=1}^{N} \log\left(p_{j,n}/\hat{p}_{j,n}\right)$

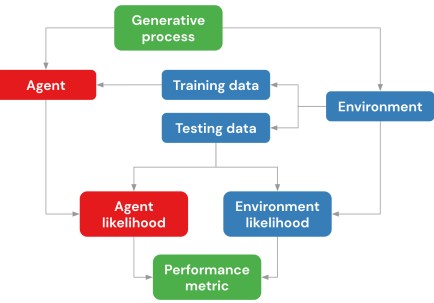

Figure 3: The Neural Testbed

## 2.1 Environment and predictions

Consider a sequence of pairs $((X_t, Y_{t+1}) : t = 0, 1, 2, \ldots)$, where each $X_t$ is a feature vector and each $Y_{t+1}$ is its target label. Each target label $Y_{t+1}$ is produced by an *environment* $\mathcal{E}$, which we formally take to be a conditional distribution $\mathcal{E}(\cdot|X_t)$. The environment $\mathcal{E}$ is a random variable; this reflects the agent's uncertainty about how labels are generated. Note that $\mathbb{P}(Y_{t+1} \in \cdot|\mathcal{E}, X_t) = \mathcal{E}(\cdot|X_t)$ and $\mathbb{P}(Y_{t+1} \in \cdot|X_t) = \mathbb{E}[\mathcal{E}(\cdot|X_t)|X_t]$.

We consider an agent that learns about the environment from training data $\mathcal{D}_T \equiv ((X_t, Y_{t+1}) : t = 0, 1, \ldots, T-1)$. After training, the agent predicts testing class labels $Y_{T+1:T+\tau} \equiv (Y_{T+1}, \ldots, Y_{T+\tau})$ from unlabeled feature vectors $X_{T:T+\tau-1} \equiv (X_T, \ldots, X_{T+\tau-1})$.

We describe the agent's predictions in terms of a generative model, parameterized by a vector $\theta_T$ that the agent learns from the training data $\mathcal{D}_T$. For any inputs $X_{T:T+\tau-1}$, $\theta_T$ determines a predictive distribution, which could be used to sample imagined outcomes $\hat{Y}_{T+1:T+\tau}$. Hence, the agents $\tau^{\text{th}}$-order predictive distribution is given by

$$\hat{P}_{T+1:T+\tau} = \mathbb{P}(\hat{Y}_{T+1:T+\tau} \in \cdot|\theta_T, X_{T:T+\tau-1}),$$

which represents an approximation to what would be obtained by conditioning on the environment:

$$P^*_{T+1:T+\tau} = \mathbb{P}\left(Y_{T+1:T+\tau} \in \cdot|\mathcal{E}, X_{T:T+\tau-1}\right).$$

If $\tau = 1$, this represents a marginal prediction; that is a prediction of a label for a single input. For $\tau > 1$, this is a joint prediction over labels for $\tau$ different inputs.

## 2.2 Kullback–Leibler loss

We use expected KL-loss to quantify the error between an agent's predictive distribution $\hat{P}_{T+1:T+\tau}$ and the prescient prediction $P^*_{T+1:T+\tau}$ that would be made given full knowledge of the environment:

$$\mathbf{d}^\tau_{\text{KL}} = \mathbb{E}\left[\mathbf{d}_{\text{KL}}\left(P^*_{T+1:T+\tau} \big\| \hat{P}_{T+1:T+\tau}\right)\right]. \tag{1}$$

The expectation is taken over all random variables, including the environment $\mathcal{E}$, the parameters $\theta_T$, $X_{T:T+\tau-1}$, and $Y_{T+1:T+\tau}$. Note that $\mathbf{d}^\tau_{\text{KL}}$ is equivalent to the widely used notion of cross-entropy loss, though offset by a quantity that is independent of $\theta_T$.

In contexts we will consider, it is not possible to compute $\mathbf{d}^\tau_{\text{KL}}$ exactly. As such, we will approximate $\mathbf{d}^\tau_{\text{KL}}$ via Monte Carlo simulation, as described by Algorithm 1. First, a set of environments is sampled. Then, for each sampled environment, a training dataset is sampled. For sampled environment and corresponding training data set, the agent is re-initialized, trained, and then tested on $N$ independent test data $\tau$-samples. Note that each test data $\tau$-sample includes $\tau$ data pairs. For each test data $\tau$-sample, the likelihood of the environment $p_{j,n}$ is computed exactly, but that of the agent's predictive distribution is approximated via another Monte Carlo simulation, and we use $\hat{p}_{j,n}$ to denote this approximation. The estimate of $\mathbf{d}^\tau_{\text{KL}}$ is taken to be the sample mean of these log-likelihood ratios.

## 3 The Neural Testbed

In this section we introduce the Neural Testbed. We believe that a simple, clear and accessible testbed can provide significant value to community. We provide a high-level overview of the open-source code which we release in Appendix A. We then provide more details on the underlying generative model, together with an extensive selection of benchmark agents that we have tuned to perform well in this setting.

### 3.1 Synthetic data generating processes

By data generating process, we do not mean only the conditional distribution of data pairs $(X_t, Y_{t+1})|\mathcal{E}$ but also the distribution of the environment $\mathcal{E}$. The Testbed considers 2-dimensional inputs and binary classification problems. The logits are sampled from a 2-hidden-layer ReLU MLP with (50,50) hidden units and Xavier initialization (Glorot and Bengio, 2010). We choose this process to be maximally simple and canonical in the deep learning world. However, we will go on to show that the key findings of this paper are not particularly sensitive to the exact choice of generative model.

The Neural Testbed estimates KL-loss, with $\tau \in \{1, 10\}$, for three temperature settings and several training dataset sizes. The temperature $\rho$ controls the signal to noise ratio as the class probabilities are given by softmax(logits/$\rho$). For each value of $\tau$, the KL-losses are averaged to produce an aggregate performance measure. Further details concerning data generation and agent evaluation are offered in Appendix B.

### 3.2 Why do we need a synthetic testbed?

The Neural Testbed is designed to be a maximally simple problem that investigates the key properties of uncertainty modeling in deep learning. Progress in deep learning has been driven both by challenge datasets that stretch agent capabilities (Deng et al., 2009; Krizhevsky et al., 2012), together with foundational work that builds understanding (Bartlett et al., 2021). In this work, we provide a benchmark designed to improve our *understanding* of probabilistic predictions beyond marginals. Doing well in the testbed is not necessarily an impressive grand success in AI, although doing poorly in such a simple setting may reveal fundamental flaws in algorithm design.

A key property of the testbed is that it is specified by a probabilistic model, rather than a finite collection of datasets. Benchmarks that rank performance on datasets are vulnerable to overfitting through iterative hill-climbing on the data included in the benchmark (Russo and Zou, 2016), which may not generalize to data outside of the benchmark (Recht et al., 2018). In contrast, access to a generative model means that we can produce an unlimited amount of testing data from our problem of interest. We can avoid the dangers of overfitting to any specific choices of benchmark dataset simply by generating more samples.

### 3.3 Benchmark agents

Table 1 lists agents that we study and compare as well as hyperparameters that we tune. In our experiments, we optimize these hyperparameters via grid search. Our implementations, which aim to match 'canonical' versions, are available in Appendix A.

In addition to these agent implementations, our open-source offerings include all the evaluation code to reproduce the results of this paper. Our experiments make extensive use of parallel computation to facilitate hyperparameter sweeps. Nevertheless, the overall computational cost is relatively low by modern deep learning standards and relies only on standard CPUs. For reference, evaluating the `mlp` agent across all the problems in our testbed requires less than 3 CPU-hours. We view our open-source effort as a substantial contribution of this work.

## 4 Results

We evaluate the benchmark agents of Section 3.3 across the Neural Testbed. We begin with an analysis of marginal predictions where, after agent tuning, all approaches are able to make reasonably good predictions. However, when we examine *joint* predictions we find that agent performance can vary drastically, even for well-tuned agents. If an agent cannot

Table 1: Summary of benchmark agents, full details in Appendix C.

| agent | description | hyperparameters |
|---|---|---|
| mlp | Vanilla MLP | $L_2$ decay |
| ensemble | 'Deep Ensemble' (Lakshminarayanan et al., 2017) | $L_2$ decay, ensemble size |
| dropout | Dropout (Gal and Ghahramani, 2016) | $L_2$ decay, network, dropout rate |
| bbb | Bayes by Backprop (Blundell et al., 2015) | prior mixture, network, early stopping |
| hypermodel | Hypermodel (Dwaracherla et al., 2020) | $L_2$ decay, prior, bootstrap, index dimension |
| ensemble+ | Ensemble + prior functions (Osband et al., 2018) | $L_2$ decay, ensemble size, prior scale, bootstrap |
| sgmcmc | Stochastic Langevin MCMC (Welling and Teh, 2011) | learning rate, prior, momentum |

output accurate joint predictions in the testbed, we should question if we expect that same agent to perform better other settings. These results provide significant new insights to the the design of effective learning agents, and are a major contribution of this paper.

## 4.1 Performance in marginal predictions

We begin our evaluation of benchmark approaches to Bayesian deep learning in marginal predictions ($\tau = 1$). One of the first questions one might consider is whether the generative model as outlined in Section 3.1 represents a meaningful challenge for deep learning systems. Figure 4 compares the performance of naive uniform class probabilities, logistic regression, and a tuned 2-layer MLP. This simple comparison demonstrates that the Neural Testbed is not trivially solved by agents without deep learning architectures.

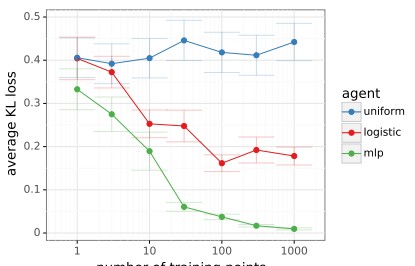

Figure 4: Performance with growing data.

| AGENT | ACCURACY | ECE | $\mathbf{d}_{\text{KL}}^1$ | $\mathbf{d}_{\text{KL}}^{10}$ |
|---|---|---|---|---|
| MLP | 0.793 | 0.078 | 0.129 | 1.367 |
| ENSEMBLE | 0.792 | 0.079 | 0.128 | 1.356 |
| DROPOUT | 0.793 | 0.080 | 0.128 | 1.347 |
| BBB | 0.792 | 0.079 | 0.129 | 1.375 |
| HYPERMODEL | 0.793 | 0.081 | 0.130 | **1.107** |
| ENSEMBLE+ | 0.790 | 0.085 | 0.129 | **1.015** |
| SGMCMC | 0.796 | 0.082 | 0.122 | **0.947** |

Table 2: Agent performance, deviation from MLP greater than 2 stderr in bold.

Marginal predictions have been the focus of the Bayesian deep learning literature. Despite this focus, Figure 2 shows that none of the benchmark methods significantly outperform a well-tuned MLP baseline in terms of $\mathbf{d}_{\text{KL}}^1$. This observation is mirrored when we examine the average classification accuracy *and* expected calibration error (ECE) across the testbed (Table 2). These results are different from the empirical observations in other challenge datasets, where much agent development has focused on improving ECE, and present an interesting new observation in the Bayesian deep learning literature (Nado et al., 2021). We have two main hypothesis for this discrepancy. First, our agents are tuned for $\mathbf{d}_{\text{KL}}^{\text{agg}} = \mathbf{d}_{\text{KL}}^1 + \frac{1}{10}\mathbf{d}_{\text{KL}}^{10}$, not ECE (see Appendix C). Second, the generative model of Section 3.1 matches the agent architecture, with inputs sampled i.i.d. $N(0, I)$. Investigating the conditions in which these results hold more generally is an exciting area for future research.

## 4.2 Performance beyond marginals

One of the key contributions of this paper is to evaluate predictive distributions beyond marginals. Figure 2 shows that sgmcmc is the top-performing agent overall. This should be reassuring to the Bayesian deep learning community and beyond. In the limit of large compute this agent should recover the 'gold-standard' of Bayesian inference, and it does indeed perform best (Welling and Teh, 2011). However, some of the most popular approaches in this field (ensemble, dropout) do not actually provide good approximations to the predictive distributions of order $\tau = 10$. In fact, we even see that ensemble+ and hypermodels can provide much better approximations to the Bayesian posterior than 'fully Bayesian' VI approaches like bbb (Wilson and Izmailov, 2020). We note too that while sgmcmc performs best, it also requires orders of magnitude more computation than competitive methods even in this toy setting (see Appendix D.3). As we scale to more complex environments, it may therefore be worthwhile to consider alternative approaches.

To see where some agents are able to outperform, we compare `ensemble` and `ensemble+` under the medium SNR regime. These agents are identical, except for the addition of a randomized prior function (Osband et al., 2018). Figure 5 shows that, although these methods perform similarly in the quality of their marginal predictions ($\tau = 1$), the addition of a prior function greatly improves the quality of joint predictive distributions ($\tau = 10$) in the low data regime. Note that, since the testbed considers 2D inputs, 100 training points may already be considered as in the high data regime. Figure 6 provides some insight for how this benefit scales with the order $\tau$ of the predictive distribution. We can see a clear trend that as $\tau$ increases so does the separation between agents `ensemble` and `ensemble+`. For more intuition on *how* prior functions are able to drive this benefit, see Appendix D.1.

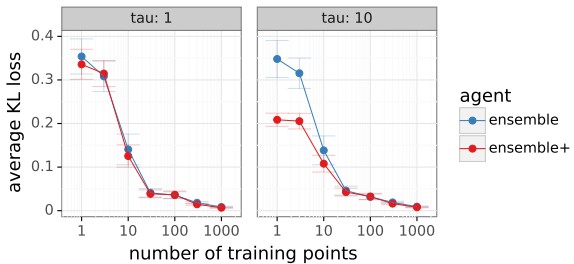

Figure 5: Prior functions help with joint predictions.

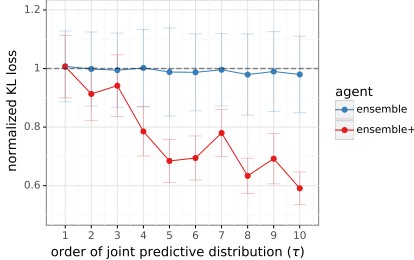

Figure 6: Benefit grows with $\tau$.

## 5 Sequential decisions

In this section, we will form a sequential decision problem based on the Neural Testbed, and show that it is the quality in *joint* predictions that is essential to driving good performance in sequential decision making. Further, we show that the insights gained from the simple 2D Neural Testbed can extend to high-dimensional decision problems.

### 5.1 Neural bandit

We use the generative model of the Neural Testbed to define a class of bandit problems (Gittins, 1979). First, we sample a set of $N$ actions $\mathcal{X} = \{x_1, \ldots, x_N\}$ i.i.d. from a $d$-dimensional standard normal distribution. We then sample an environment $\mathcal{E}$, which specifies the conditional probability $\mathcal{E}(Y_{t+1} \in \cdot | X_t)$, according to the class of generative models described in Section 3.1. We pick the temperature, which controls the SNR, to be 0.1. At each timestep $t$, an agent selects an action $X_t \in \mathcal{X}$ and receives a reward $R_{t+1} = Y_{t+1}$. Let $\overline{R}_x = \mathbb{E}[R_{t+1} | \mathcal{E}, X_t = x]$ denote the expected reward of action $x$ conditioned on the environment, and let $X_* = \arg\max_{x \in \mathcal{X}} \overline{R}_x$ denote the optimal action. We assess agent performance through $\texttt{regret}(T) := \sum_{t=0}^{T-1} \mathbb{E}\left[\overline{R}_{X_*} - \overline{R}_{X_t}\right]$, which measures the shortfall in expected cumulative rewards relative to an optimal decision maker.

We evaluate the testbed agents on these bandit problems through actions selected by Thompson sampling, varying only the posterior predictive distributions that TS samples from. A TS agent requires an approximate posterior distribution over the environment, which is supplied by the testbed agents. At each timestep, TS samples an environment from the approximate posterior and selects an action that optimizes for the sampled environment (Thompson, 1933; Russo et al., 2018). A complete algorithm is presented in Appendix E.

### 5.2 Agent performance

We present empirical results of testbed agents on these random bandit problems with $N = 1000$ actions drawn from a $d = 50$ dimensional space. Figure 7 shows the average regret through time for each of the agents as selected by the Neural Testbed, averaged over 20 random seeds.[2] We can see that for each learning agent, the quality of decisions improves through time. However, the quality of decisions is greatly affected by the choice of agent.

---

[2] We omit `sgmcmc` as the computational demands are several orders of magnitude too large to consider in online learning.

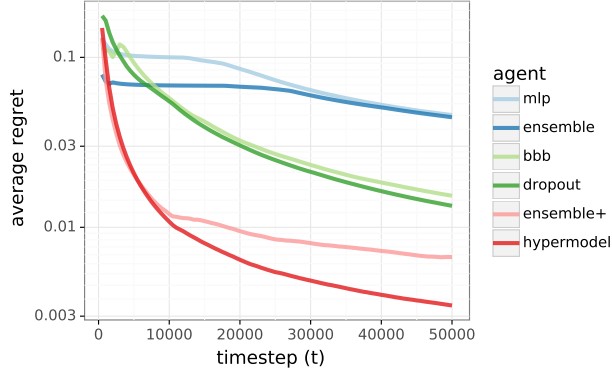

Figure 7: Learning agent impacts TS regret in neural bandits.

To investigate the relationship between predictions and decisions we repeat the experiment of Figure 7 with 10 independent random initializations over all the testbed and bandit problems. We then empirically investigate the correlation between $\mathbf{d}_{\mathrm{KL}}^{\tau}$ and total regret at $T = 50,000$ for both $\tau = 1$ and $\tau = 10$. We use bootstrap sampling to estimate confidence intervals on the correlation coefficient on a logarithmic scale at the 5th and 95th percentiles. Figures 8 and 9 support our claim that performance in $\mathbf{d}_{\mathrm{KL}}^{10}$ is highly correlated with performance in sequential decision problems, whereas correlation to marginals is not significant. We would not expect a perfect correlation as the particular TS action selection strategy may introduce confounding factors, together with natural variability in seeds.

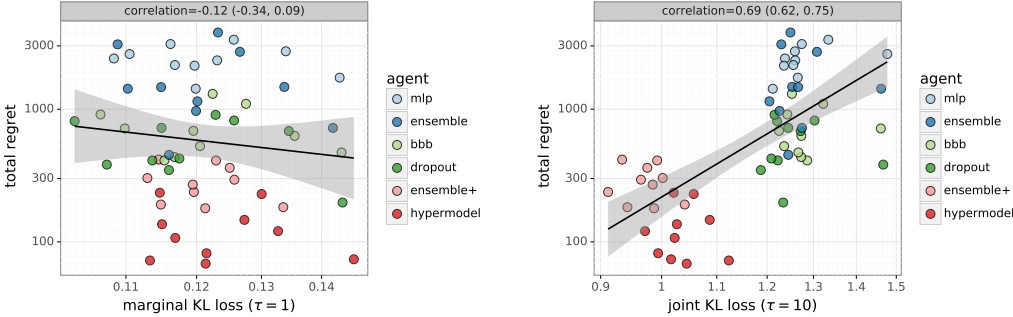

Figure 8: Testbed marginal performance is not significantly correlated with regret.

Figure 9: Testbed joint performance is highly correlated with regret.

## 6 Robustness of generative model

The experiments of Sections 4 and 5 are all performed with the generative model as described in Section 3.1. One natural concern is that these results might be sensitive to this choice of model, and so be less transferable to general deep learning research. In this section we repeat these analyses under different generative models. We find that the quality of *joint* predictions and bandit performance is extremely robust across choice of generative models.

For these experiments we take the tuned agents of Section 4 and then evaluate these agents under different generative models. Whereas these agent hyperparameters were tuned for the 2-layer ReLU MLP with 50-50 hidden units, we will also these agents over alternative environments varying:

- **activation**=[tanh, swish, sigmoid, selu, relu, leaky relu, gelu] (Figure 10).
- **hidden units**=[5, 10, 20, 50, 100] (Figure 11).

Evaluation for each of these environments $\mathcal{E}_i$ proceeds as before: the agent is trained on data generated by $\mathcal{E}_i$ and then evaluated on the quality of predictions on testing data from $\mathcal{E}_i$. If the qualitative results under different environments are similar, then we know that our results are somewhat robust to the exact generative model we choose.

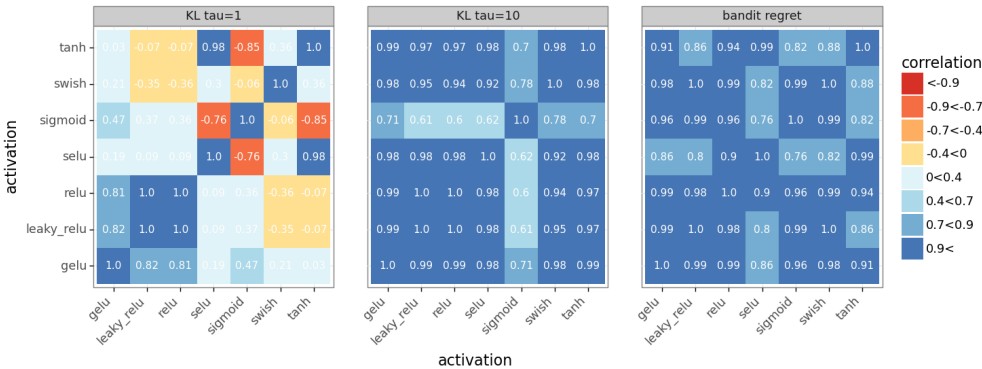

Figure 10: Correlation of agent performance across different activation functions.

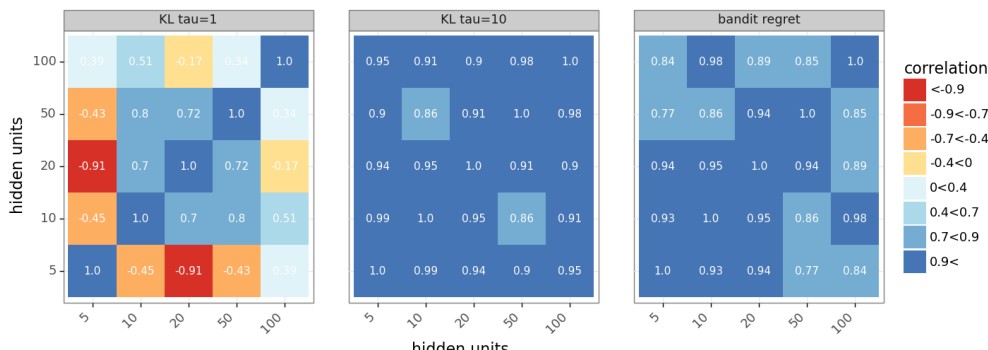

Figure 11: Correlation of agent performance across different hidden units.

Figure 10 and 11 examine the empirical correlation coefficient between the vector of agent evaluations, under the metrics $\mathbf{d}_{\mathrm{KL}}^1, \mathbf{d}_{\mathrm{KL}}^{10}$ and bandit regret. We see that, the marginal evaluations are highly correlated for 'similar' generative models (e.g. ReLU and leaky ReLU) but can even be anti-correlated when the models stray too far. However, the correlations are very high across a wide range of generative models when we look at either the quality of joint predictions or the regret in the bandit problems. These results help to build confidence in the key observations we make in this paper. Notably, they suggest that the separation of agents in terms of performance on joint prediction (Figure 2) is not too sensitive to the choice of generative model, and so may hold some wider insight relevant to the community. Follow up work has confirmed that these results are also highly correlated with performance on benchmark datasets (Osband et al., 2022).

## 7 Conclusion

The Neural Testbed investigates the quality of predictive uncertainty in joint predictions, as well as marginals. With this simple and clear 2D challenge we aim to build understanding that can inform the field's wider efforts in deep learning. We have shown that results on the testbed can offer new insights to agent development. Further, we establish that the insights gained in the testbed can scale up to complex and high-dimensional decision problems.

Beyond the results in this paper, we believe this work can provide a base for future research:

- Can we design better learning algorithms for joint predictions, as well as marginals?
- Are there analogous results to Figure 2 on large-scale challenge datasets?
- How can effective joint predictions drive better decisions?

We believe that studying these simple testbed problems can help foster interplay between theory and practice, improve accessibility in the field, and complement existing research. We hope that this will accelerate the growth of *understanding* in the field and, ultimately, drive forward the design of better learning agents.

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
