# A  Open source code

This section is meant to give an overview of our opensource code. Together with our paper submission we include a link to anonymous github repository.

- `neural_testbed`: https://github.com/deepmind/neural_testbed

Together with this git repo, we include a 'tutorial colab' – a Jupyter notebooks that can be run in the browser without requiring any local installation at `neural_testbed/tutorial.ipynb`. Our library is written in Python, and relies heavily on JAX for scientific computing (Bradbury et al., 2018). We view this open-source effort as a major contribution of our paper.

# B  Testbed Pseudocode

We present the testbed pseudocode in this section. Specifically, Algorithm 2 is the pseudocode for our neural testbed, and Algorithm 3 is an approach to estimate the likelihood of a test data $\tau$-sample conditioned on an agent's belief, based on the standard Monte-Carlo estimation. The presented testbed pseudocode works for any prior $\mathbb{P}(\mathcal{E} \in \cdot)$ over the environment and any input distribution $P_X$, including the ones described in Section 3.1. We also release full code and implementations in Appendix A.

In addition to presenting the testbed pseudocode, we also explain our choices of experiment parameters in Appendix C. To apply Algorithm 2, we need to specify an input distribution $P_X$ and a prior distribution on the environment $\mathbb{P}(\mathcal{E} \in \cdot)$. Recall from Section 3.1 that we consider binary classification problems with input dimension 2. We choose $P_X = N(0, I)$, and we consider three environment priors distinguished by a temperature parameter that controls the signal-to-noise ratio (SNR) regime. We sweep over temperatures in $\{0.01, 0.1, 0.5\}$. The prior distribution $\mathbb{P}(\mathcal{E} \in \cdot)$ is induced by a distribution over MLPs with 2 hidden layers and ReLU activation. The MLP is distributed according to standard Xavier initialization, except that biases in the first layer are drawn from $N(0, \frac{1}{2})$. The MLP outputs two units, which are divided by the temperature parameter and passed through the softmax function to produce class probabilities. The implementation of this generative model is in our open source code under the path `/generative/factories.py`.

We now describe the other parameters we use in the Testbed. In Algorithm 2, we pick the order of predictive distributions $\tau \in \{1, 10\}$, training dataset size $T \in \{1, 3, 10, 30, 100, 300, 1000\}$, number of sampled problems $J = 10$, and number of testing data $\tau$-samples $N = 1000$. To apply Algorithm 3, we sample $M = 1000$ models from the agent.

# C  Agents

In this section, we describe the benchmark agents in Section 3.3 and the choice of various hyperparameters used in the implementation of these agents. The list of agents include MLP, ensemble, dropout, Bayes by backprop, stochastic Langevin MCMC, ensemble+ and hypermodel. We will also include other agents such as KNN, random forest, and deep kernel, but the performance of these agents was worse than the other benchmark agents, so we chose not to include them in the comparison in Section 4. In each case, we attempt to match the "canonical" implementation. The complete implementation of these agents including the hyperparameter sweeps used for the Testbed are available in Appendix A. We make use of the Epistemic Neural Networks notation from (Osband et al., 2021) in our code. We set the default hyperparameters of each agent to be the ones that minimize the aggregated KL score $\mathbf{d}_{\mathrm{KL}}^{\mathrm{agg}} = \mathbf{d}_{\mathrm{KL}}^{1} + \frac{1}{10}\mathbf{d}_{\mathrm{KL}}^{10}$.

## C.1  MLP

The `mlp` agent learns a 2-layer MLP with 50 hidden units in each layer by minimizing the cross-entropy loss with $L_2$ weight regularization. The $L_2$ weight decay scale is chosen either to be $\lambda\frac{1}{T}$ or $\lambda\frac{d\sqrt{\beta}}{T}$, where $d$ is the input dimension, $\beta$ is the temperature of the generative process and

---
**Algorithm 2** Neural Testbed
---

**Require:** the testbed requires the following inputs

      1. prior distribution over the environment $\mathbb{P}(\mathcal{E} \in \cdot)$, input distribution $P_X$

      2. agent $f_\theta$

      3. number of training data $T$, test distribution order $\tau$

      4. number of sampled problems $J$, number of test data samples $N$

      5. parameters for agent likelihood estimation, as is specified in Algorithm 3

**for** $j = 1, 2, \ldots, J$ **do**

  **Step 1: sample environment and training data**

      1. sample environment $\mathcal{E} \sim \mathbb{P}(\mathcal{E} \in \cdot)$

      2. sample $T$ inputs $X_0, X_1, \ldots, X_{T-1}$ i.i.d. from $P_X$

      3. sample the training labels $Y_1, \ldots, Y_T$ conditionally i.i.d. as

$$Y_{t+1} \sim \mathbb{P}\left(Y \in \cdot | \mathcal{E}, X = X_t\right) \quad \forall t = 0, 1, \ldots, T-1$$

      4. choose the training dataset as $\mathcal{D}_T = \{(X_t, Y_{t+1}), t = 0, \ldots, T-1\}$

  **Step 2: train agent**

    train agent $f_{\theta_T}$ based on training dataset $\mathcal{D}_T$

  **Step 3: compute likelihoods**

    **for** $n = 1, 2, \ldots, N$ **do**

      1. sample $X_T^{(n)}, \ldots, X_{T+\tau-1}^{(n)}$ i.i.d. from $P_X$

      2. generate $Y_{T+1}^{(n)}, \ldots, Y_{T+\tau}^{(n)}$ conditionally independently as

$$Y_{t+1}^{(n)} \sim \mathbb{P}\left(Y \in \cdot \Big| \mathcal{E}, X = X_t^{(n)}\right) \quad \forall t = T, T+1, \ldots, T+\tau-1$$

      3. compute the likelihood under the environment $\mathcal{E}$ as

$$p_{j,n} = \mathbb{P}\left(Y_{T+1:T+\tau}^{(n)} \Big| \mathcal{E}, X_{T:T+\tau-1}^{(n)}\right) = \prod_{t=T}^{T+\tau-1} \Pr\left(Y_{t+1}^{(n)} \Big| \mathcal{E}, X_t^{(n)}\right)$$

      4. estimate the likelihood conditioned on the agent's belief

$$\hat{p}_{j,n} \approx \mathbb{P}\left(\hat{Y}_{T+1:T+\tau} = Y_{T+1:T+\tau}^{(n)} \Big| \theta_T, X_{T:T+\tau-1}^{(n)}, Y_{T+1:T+\tau}^{(n)}\right),$$

      based on Algorithm 3 with test data $\tau$-sample $\left(X_{T:T+\tau-1}^{(n)}, Y_{T+1:T+\tau}^{(n)}\right)$.

  **end for**

  **return** $\frac{1}{JN} \sum_{j=1}^{J} \sum_{n=1}^{N} \log\left(p_{j,n}/\hat{p}_{j,n}\right)$

---

---
**Algorithm 3** Monte Carlo Estimation of Likelihood of Agent's Belief
---

**Require:** the Monte-Carlo estimation requires the following inputs

      1. trained agent $f_{\theta_T}$ and number of Monte Carlo samples $M$

      2. test data $\tau$-sample $(X_{T:T+\tau-1}, Y_{T+1:T+\tau})$

**Step 1:** sample $M$ models $\hat{\mathcal{E}}_1, \ldots, \hat{\mathcal{E}}_M$ conditionally i.i.d. from $\mathbb{P}\left(\hat{\mathcal{E}} \in \cdot \Big| \theta_T\right)$

**Step 2:** estimate $\hat{p}$ as

$$\hat{p} = \frac{1}{M} \sum_{m=1}^{M} \mathbb{P}\left(\hat{Y}_{T+1:T+\tau} = Y_{T+1:T+\tau} \Big| \hat{\mathcal{E}}_m, X_{T:T+\tau-1}, Y_{T+1:T+\tau}\right)$$

**return** $\hat{p}$

---

$T$ is the size of the training dataset. We sweep over $\lambda \in \{10^{-4}, 10^{-3}, 10^{-2}, 10^{-1}, 1, 10, 100\}$. We implement the MLP agent as a special case of a deep ensemble (C.2). The implementation and hyperparameter sweeps for the `mlp` agent can be found in our open source code, as a special case of the `ensemble` agent, under the path `/agents/factories/ensemble.py`.

## C.2  Ensemble

We implement the basic "deep ensembles" approach for posterior approximation (Lakshminarayanan et al., 2017). The `ensemble` agent learns an ensemble of MLPs by minimizing the cross-entropy loss with $L_2$ weight regularization. The only difference between the ensemble members is their independently initialized network weights. We chose the $L_2$ weight scale to be either $\lambda \frac{1}{MT}$ or $\lambda \frac{d\sqrt{\beta}}{MT}$, where $M$ is the ensemble size, $d$ is the input dimension, $\beta$ is the temperature of the generative process, and $T$ is the size of the training dataset. We sweep over ensemble size $M \in \{1, 3, 10, 30, 100\}$ and $\lambda \in \{10^{-4}, 10^{-3}, 10^{-2}, 10^{-1}, 1, 10, 100\}$. We find that larger ensembles work better, but this effect is within margin of error after 10 elements. The implementation and hyperparameter sweeps for the `ensemble` agent can be found in our open source code under the path `/agents/factories/ensemble.py`.

## C.3  Dropout

We follow Gal and Ghahramani (2016) to build a `droput` agent for posterior approximation. The agent applies dropout on each layer of a fully connected MLP with ReLU activation and optimizes the network using the cross-entropy loss combined with $L_2$ weight decay. The $L_2$ weight decay scale is chosen to be either $\frac{l^2}{2T}(1 - p_{\text{drop}})$ or $\frac{d\sqrt{\beta}l}{T}$ where $p_{\text{drop}}$ is the dropping probability, $d$ is the input dimension, $\beta$ is the temperature of the data generating process, and $T$ is the size of the training dataset. We sweep over dropout rate $p_{\text{drop}} \in \{0, 0.1, 0.2, 0.3, 0.4, 0.5, 0.6, 0.7, 0.8, 0.9\}$, length scale (used for $L_2$ weight decay) $l \in \{1, 3, 10\}$, number of neural network layers $\in \{2, 3\}$, and hidden layer size $\in \{50, 100\}$. The implementation and hyperparameter sweeps for the `dropout` agent can be found in our open source code under the path `/agents/factories/dropout.py`.

## C.4  Bayes-by-backprop

We follow Blundell et al. (2015) to build a `bbb` agent for posterior approximation. We consider a scale mixture of two zero-mean Gaussian densities as the prior. The Gaussian densities have standard deviations $\sigma_1$ and $\sigma_2$, and they are mixed with probabilities $p$ and $1 - p$, respectively. We sweep over $\sigma_1 \in \{0.3, 0.5, 0.7, 1, 2, 4\}$, $\sigma_2 \in \{0.3, 0.5, 0.7\}$, $p \in \{0, 0.5, 1\}$, learning rate $\in \{10^{-3}, 3 \times 10^{-3}\}$, number of training steps $\in \{1000, 2000\}$, number of neural network layers $\in \{2, 3\}$, hidden layer size $\in \{50, 100\}$, and the ratio of the complexity cost to the likelihood cost $\in \{1, d\sqrt{\beta}\}$, where $d$ is the input dimension and $\beta$ is the temperature of the data generating process. The implementation and hyperparameter sweeps for the `bbb` agent can be found in our open source code under the path `/agents/factories/bbb.py`.

## C.5  Stochastic gradient Langevin dynamics

We follow Welling and Teh (2011) to implement a `sgmcmc` agent using stochastic gradient Langevin dynamics (SGLD). We consider two versions of SGLD, one with momentum and other without the momentum. We consider independent Gaussian prior on the neural network parameters where the prior variance is set to be

$$\sigma^2 = \lambda \frac{T}{d\sqrt{\beta}},$$

where $\lambda$ is a hyperparameter that is swept over $\{0.0025, 0.01, 0.04\}$, $d$ is the input dimension, $\beta$ is the temperature of the data generating process, and $T$ is the size of the training dataset. We consider a constant learning rate that is swept over $\{10^{-4}, 5 \times 10^{-4}, 10^{-3}, 5 \times 10^{-3}\}$. For SGLD with momentum, the momentum decay term is always set to be 0.9. The number of training batches is $5 \times 10^5$ with burn-in time of $10^5$ training batches. We save a model every 1000 steps after the burn-in time and use these models as an ensemble during the evaluation.

The implementation and hyperparameter sweeps for the `sgmcmc` agent can be found in our open source code under the path `/agents/factories/sgmcmc.py`.

## C.6  Ensemble+

We implement the `ensemble+` agent using deep ensembles with randomized prior functions (Osband et al., 2018) and bootstrap sampling (Osband and Van Roy, 2015). Similar to the vanilla ensemble agent in Section C.2, we consider $L_2$ weight scale to be either $\lambda\frac{1}{MT}$ or $\lambda\frac{d\sqrt{\beta}}{MT}$. We sweep over ensemble size $M \in \{1, 3, 10, 30, 100\}$ and $\lambda \in \{0.1, 0.3, 1, 3, 10\}$. The randomized prior functions are sampled exactly from the data generating process, and we use a prior scale of $3/\sqrt{\beta}$. In addition, we sweep over bootstrap type (none, exponential, bernoulli).

Note that an ensemble+ agent is obtained by an addition of a prior network to the ensemble agent. We find that the addition of randomized prior functions is crucial for improvement in performance over vanilla deep ensembles in terms of the quality of joint predictions. The implementation and hyperparameter sweeps for the `ensemble+` agent can be found in our open source code under the path `/agents/factories/ensemble_plus.py`.

## C.7  Hypermodel

We follow Dwaracherla et al. (2020) to build a `hypermodel` agent for posterior approximation. We consider a linear hypermodel over a 2-layer MLP base model. We sweep over index dimension $\in \{1, 3, 5, 7\}$. The $L_2$ weight decay is chosen to be either $\lambda\frac{1}{T}$ or $\lambda\frac{d\sqrt{\beta}}{T}$ with $\lambda \in \{0.1, 0.3, 1, 3, 10\}$, where $d$ is the input dimension, $\beta$ is the temperature of the data generating process, and $T$ is the size of the training dataset. We sweep over bootstrap type (none, exponential, bernoulli). We use an additive prior which is a linear hypermodel prior over an MLP base model, which is similar to the generating process, with number of hidden layers in $\{1, 2\}$, 10 hidden units in each layer, and prior scale from $\{1/\sqrt{\beta}, 1/\beta\}$. The implementation and hyperparameter sweeps for the `hypermodel` agent can be found in our open source code under the path `/agents/factories/hypermodel.py`.

## C.8  Non-parametric classifiers

K-nearest neighbors (k-NN) (Cover and Hart, 1967) and random forest classifiers (Friedman, 2017) are simple and cheap off-the-shelf non-parametric baselines (Murphy, 2012; Pedregosa et al., 2011). The 'uncertainty' in these classifiers arises merely from the fact that they produce distributions over the labels and as such we do not expect them to perform well relative to more principled approaches. Moreover, these methods have no capacity to model $\mathbf{d}_{\text{KL}}^{\tau}$ for $\tau > 1$. For the `knn` agent we swept over the number of neighbors $k \in \{1, 5, 10, 30, 50, 100\}$ and the weighting of the contribution of each neighbor as either uniform or based on distance. For the `random_forest` agent we swept over the number of trees in the forest $\{10, 100, 1000\}$, and the splitting criterion which was either the Gini impurity coefficient or the information gain. To prevent infinite values in the KL we truncate the probabilities produced by these classifiers to be in the interval $[0.01, 0.99]$. The implementation and hyperparameter sweeps for the `knn` and `random_forest` agents can be found in our open source code under the paths `/agents/factories/knn.py` and `/agents/factories/random_forest.py`.

## C.9  Gaussian process with learned kernel

A neural network takes input $X_t \in \mathcal{X}$ and produces output $Z_{t+1} = W\phi_\theta(X_t) + b \in \mathbf{R}^K$, where $W \in \mathbf{R}^{K \times m}$ is a matrix, $b \in \mathbb{R}^K$ is a bias vector, and $\phi_\theta : \mathcal{X} \to \mathbb{R}^m$ is the output of the penultimate layer of the neural network. In the case of classification the output $Z_{t+1}$ corresponds to the logits over the class labels, i.e., $\hat{Y}_{t+1} \propto \exp(Z_{t+1})$. The neural network should learn a function that maps the input into a space where the classes are linearly distinguishable. In other words, the mapping that the neural network is learning can be considered a form of *kernel* (Schölkopf and Smola, 2018), where the kernel function $k : \mathcal{X} \times \mathcal{X} \to \mathbf{R}$ is simply $k(X, X') = \phi_\theta(X)^\top \phi_\theta(X')$. With this in mind, we can take a *trained*

neural network and consider the learned mapping to be the kernel in a Gaussian process (GP) (Rasmussen, 2003), from which we can obtain approximate uncertainty estimates. Concretely, let $\Phi_{0:T-1} \in \mathbf{R}^{T \times m}$ be the matrix corresponding to the $\phi_\theta(X_t)$, $t = 0, \ldots, T-1$, vectors stacked row-wise and let $\Phi_{T:T+\tau-1} \in \mathbf{R}^{\tau \times m}$ denote the same quantity for the test set. We can write the kernel function evaluated on the training and test datasets using these matrices. Fix index $i \in \{0, \ldots, K-1\}$ to be a particular class index. A GP models the joint distribution over the dataset to be a multi-variate Gaussian, i.e.,

$$\begin{bmatrix} Z_{1:T}^{(i)} \\ Z_{T+1:T+\tau}^{(i)} \end{bmatrix} \sim \mathcal{N}\left(\begin{bmatrix} \mu_{1:T}^{(i)} \\ \mu_{T+1:T+\tau}^{(i)} \end{bmatrix}, \begin{bmatrix} \sigma^2 I + \Phi_{0:T-1}\Phi_{0:T-1}^\top & \Phi_{T:T+\tau-1}\Phi_{0:T-1}^\top \\ \Phi_{0:T-1}\Phi_{T:T+\tau-1}^\top & \Phi_{T:T+\tau-1}\Phi_{T:T+\tau-1}^\top \end{bmatrix}\right)$$

where $\sigma > 0$ models the noise in the training data measurement and $\mu_{1:T}^{(i)}$, $\mu_{T+1:T+\tau}^{(i)}$ are the means under the GP. The conditional distribution is given by

$$P(Z_{T+1:T+\tau}^{(i)} \mid Z_{1:T}^{(i)}, X_{0:T+\tau-1}) = \mathcal{N}\left(\mu_{T+1:T+\tau|1:T}^{(i)}, \Sigma_{T+1:T+\tau|1:T}\right)$$

where

$$\Sigma_{T+1:T+\tau|1:T} = \Phi_{T:T+\tau-1}\Phi_{T:T+\tau-1}^\top - \Phi_{T:T+\tau-1}\Phi_{0:T-1}^\top(\sigma^2 I + \Phi_{0:T-1}\Phi_{0:T-1}^\top)^{-1}\Phi_{0:T-1}\Phi_{T:T+\tau-1}^\top.$$

and rather than use the GP to compute $\mu_{T+1:T+\tau|0:T}^{(i)}$ (which would not be possible since we do not observe the true logits) we just take it to be the output of the neural network when evaluated on the test dataset. The matrix being inverted in the expression for $\Sigma_{T+1:T+\tau|0:T}$ has dimension $T \times T$, which may be quite large. We use the Sherman-Morrison-Woodbury identity to rewrite it as follows (Woodbury, 1950)

$$\Sigma_{T+1:T+\tau|0:T} = \Phi_{T:T+\tau-1}(I - \Phi_{0:T-1}^\top(\sigma^2 I + \Phi_{0:T-1}\Phi_{0:T-1}^\top)^{-1}\Phi_{0:T-1})\Phi_{T:T+\tau-1}^\top$$
$$= \sigma^2 \Phi_{T:T+\tau-1}(\sigma^2 I + \Phi_{0:T-1}^\top\Phi_{0:T-1})^{-1}\Phi_{T:T+\tau-1}^\top,$$

which instead involves the inverse of an $m \times m$ matrix, which may be much smaller. If we perform a Cholesky factorization of positive definite matrix $(\sigma^2 I + \Phi_{0:T-1}^\top\Phi_{0:T-1}) = LL^\top$ then the samples for all logits simultaneously can be drawn by first sampling $\zeta \in \mathbf{R}^{m \times K}$, with each entry drawn IID from $\mathcal{N}(0,1)$, then forming

$$\hat{Y}_{T+1:T+\tau} \propto \exp(\mu_{T+1:T+\tau|1:T} + \sigma\Phi_{T:T+\tau-1}L^{-\top}\zeta).$$

The implementation and hyperparameter sweeps for the `deep_kernel` agent can be found in our open source code under the path `/agents/factories/deep_kernel.py`.

## C.10 Other agents

In our paper we have made a concerted effort to include representative and canonical agents across different families of Bayesian deep learning and adjacent research. In addition to these implementations, we performed extensive tuning to make sure that each agent was given a fair shot. However, with the proliferation of research in this area, it was not possible for us to evaluate all competing approaches. We hope that, by opensourcing the Neural Testbed, we can allow researchers in the field to easily assess and compare their agents to these baselines.

For example, we highlight a few recent pieces of research that might be interesting to evaluate in our setting. Of course, there are many more methods to compare and benchmark. We leave this open as an exciting area for future research.

- **Neural Tangent Kernel Prior Functions** (He et al., 2020). Proposes a specific type of prior function in *ensemble+* inspired by connections to the neural tangent kernel.
- **Functional Variational Bayesian Neural Networks** (Sun et al., 2019). Applies variational inference directly to the function outputs, rather than weights like `bbb`.
- **Variational normalizing flows** (Rezende and Mohamed, 2015). Applies variational inference over a more expressive family than `bbb`.
- **No U-Turn Sampler** (Hoffman et al., 2014). Another approach to `sgmcmc` that attempts to compute the posterior directly, computational costs can grow large.

# D   Testbed results

In this section, we provide the complete results of the performance of benchmark agents on the Testbed, broken down by the temperature setting, which controls the SNR, and the size of the training dataset. We select the best performing agent, based on aggregated score $\mathbf{d}_{KL}^1 + \mathbf{d}_{KL}^{10}/10$, within each agent family and plot $\mathbf{d}_{KL}^1$ and $\mathbf{d}_{KL}^{10}$ with the performance of an MLP agent as a reference. We also provide a plot comparing the training time of different agents.

## D.1   Visualizing `ensemble` vs `ensemble+`

Figure 12 provides additional intuition into *how* the randomized prior functions are able to drive improved performance. Figure 12a shows a sampled generative model from our Testbed, with the training data shown in red and blue circles. Figure 12b shows the mean predictions and 4 randomly sampled ensemble members from each agent (top=`ensemble`, bottom=`ensemble+`). We see that, although the agents mostly agree in their mean predictions, `ensemble+` produces more diverse sampled outcomes enabled by the addition of randomized prior functions. In contrast, `ensemble` produces similar samples, which may explain why its performance is close to baseline `mlp` in this setting.

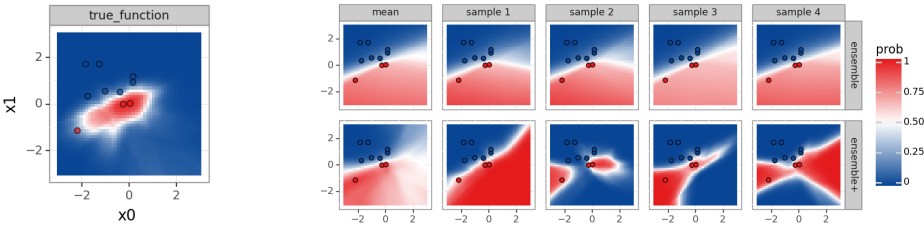

(a) True model.          (b) Agent samples: only ensemble+ produces diverse decision boundaries.

Figure 12: Visualization of the predictions of ensemble and ensemble+ agents.

## D.2   Performance breakdown

Figures 13 and 14 show the KL estimates evaluated on $\tau = 1$ and $\tau = 10$, respectively. For each agent, for each SNR regime, for each number of training points we plot the average KL estimate from the Testbed. In each plot, we include the "baseline" `mlp` agent as a black dashed line to allow for easy comparison across agents. A detailed description of these benchmark agents can be found in Appendix C.

## D.3   Training time

Figure 15 shows a plot comparing the $\mathbf{d}_{KL}^{10}$ and training time of different agents normalized with that of an MLP. The parameters of each agent are selected to maximize the $\mathbf{d}_{KL}^{10}$. We can see that `sgmcmc` agent has the best performance, but at the cost of more training time (computation). Both `ensemble+` and `hypermodel` agents have similar performance as `sgmcmc` with lower training time. We trained our agents on CPU only systems.

# E  Sequential Decision Problems

This section provides supplementary information for the sequential decision problems in Section 5. All of the code necessary to reproduce the experiments is opensourced in the `/bandit/` directory.

## E.1  Problem formulation

We consider bandit problems derived from the testbed and evaluate the agents using Algorithm 4 for which we need to specify prior on the environment $\mathbb{P}(\mathcal{E} \in \cdot)$, input distribution $P_X$, and the number of actions $N$. We choose input distribution $P_X = \mathcal{N}(0, I_d)$, where $d$ is the input dimension. We sweep over $d \in \{2, 10, 50\}$ and choose the number of actions to be $N = 20 d$, i.e., for input dimensions $\{2, 10, 50\}$ we have $\{40, 200, 1000\}$ actions respectively. We use the same prior distribution of environments as in Appendix B with a fixed temperature of 0.1. For each setting, we run for $50,000$ time steps ($T = 50,000$) and with 20 random seeds ($J = 20$).

## E.2  Agent definition

In Appendix C, we described benchmark agents in our testbed. Among these agents, we use `mlp`, `ensemble`, `dropout`, `bbb`, `ensemble+`, and `hypermodel` agents for sequential decision problems. For all the agents we use the hyper parameters specified by default, in the source code, at the path `/agents/factories/`. The default hyperparameters of each agent correspond to be the ones that minimize the aggregated KL score $\mathbf{d}_{\mathrm{KL}}^{\mathrm{agg}} = \mathbf{d}_{\mathrm{KL}}^1 + \mathbf{d}_{\mathrm{KL}}^{10}/10$. As the agent interacts with the environment, the amount of data the agent has observed keeps growing. Due to this we tune the regularization term based on the number of time steps agent has interacted with the environment. For `mlp`, `ensemble`, `ensemble+`, and `hypermodel` agents we use an $L_2$ weight decay of $\lambda \frac{2\sqrt{\beta}}{t}$, where $\beta$ is the temperature, $t$ is the number of the time steps the agent has interacted with the environment, and $\lambda$ is the default weight scale of the agent. For `dropout` we choose the $L_2$ weight decay as $\frac{2\sqrt{\beta l}}{t}$, where $l$ is the default length scale used in the dropout agent. For `bbb` we scale the prior term by $\frac{1}{t}$. As described above, all hyperparmeters are chosen to be the ones which minimize the aggregated KL score $\mathbf{d}_{\mathrm{KL}}^{\mathrm{agg}} = \mathbf{d}_{\mathrm{KL}}^1 + \frac{1}{10}\mathbf{d}_{\mathrm{KL}}^{10}$.

## E.3  Results

Figures 8 and 9 shows the correlation between performance on testbed performance and sequential decision problems with an input dimension of 50. Different points of an agent in these figures corresponds to different random seeds for the testbed and sequential problems. We can see that performance on sequential decision problems is strongly correlated with testbed joint performance $\tau = 10$ and not correlated with the testbed marginal performance. In Figures 16 and 17 we show a similar correlation plots between testbed performance and sequential decision problems across different input dimensions for sequential decision problems. We can see that performance on sequential decision problems has clear correlation with testbed joint performance $\tau = 10$, and no correlation with testbed marginal performance $\tau = 1$, across all the input dimensions considered.

These results offer empirical evidence that practical deep learning approaches separated by the quality of their joint predictions, but not their marginals, can lead to differing performance in downstream tasks. In addition, we show that our simple 2D testbed can provide insights that scale to much higher dimension problems.

---

**Algorithm 4** Evaluation on Bandit Problem

---

**Require:** Evaluation on bandit problem requires the following inputs
   1. Distribution over the environment $\mathbb{P}(\mathcal{E} \in \cdot)$, input distribution $P_X$, and the number of actions $N$.
   2. Agent $f_\theta$
   3. Number of time steps $T$
   4. Number of sampled problems $J$

**for** $j = 1, 2, \ldots, J$ **do**
   **Step 1: Sample environment and action set**
   1. Sample environment $\mathcal{E} \sim \mathbb{P}(\mathcal{E} \in \cdot)$
   2. Sample a set $\mathcal{X}$ of $N$ actions $x_1, x_2, \ldots, x_N$ i.i.d. from $P_X$
   3. Obtain the mean rewards corresponding to actions in $\mathcal{X}$ conditioned on the environment

$$\overline{R}_x = \mathbb{P}(Y_{t+1} = 1 | \mathcal{E}, X_t = x), \quad \forall x \in \mathcal{X}$$

   4. Compute the optimal expected reward $\overline{R}_* = \max_{x \in \mathcal{X}} \overline{R}_x$
   **Step 2: Agent interaction with the environment**
   Initialize the data buffer $\mathcal{D}_0 = \{\}$
   **for** $t = 1, 2, \ldots, T$ **do**
      1. Update agent $f_{\theta_t}$ belief distribution based on the data in the buffer $\mathcal{D}_{t-1}$
      2. TS action selection scheme:
         i. Sample $\hat{\mathcal{E}}_t$ from the agent belief distribution

$$\hat{\mathcal{E}}_t \sim \mathbb{P}\left(\hat{\mathcal{E}} \in \cdot | \theta_t\right)$$

         ii. Act greedily based on $\hat{\mathcal{E}}_t$

$$X_t \in \arg\max_{x \in \mathcal{X}} \mathbb{P}(\hat{Y}_{t+1} = 1 | \hat{\mathcal{E}}_t, X_t = x)$$

         iii. Generate observation $Y_{t+1}$ based on action $X_t$

$$Y_{t+1} \sim \mathbb{P}\left(Y_{t+1} \in \cdot | \mathcal{E}, X_t = X_t\right)$$

      3. Update the buffer $\mathcal{D}_t = \mathcal{D}_0 \cup (X_t, Y_{t+1})$
   **end for**
   Compute the total regret incurred in $T$ time steps

$$\text{Regret}_j(T) = \sum_{t=1}^{T} \left(\overline{R}_* - \overline{R}_{X_t}\right)$$

**end for**
**return** $\frac{1}{J} \sum_{j=1}^{J} \text{Regret}_j(T)$

---

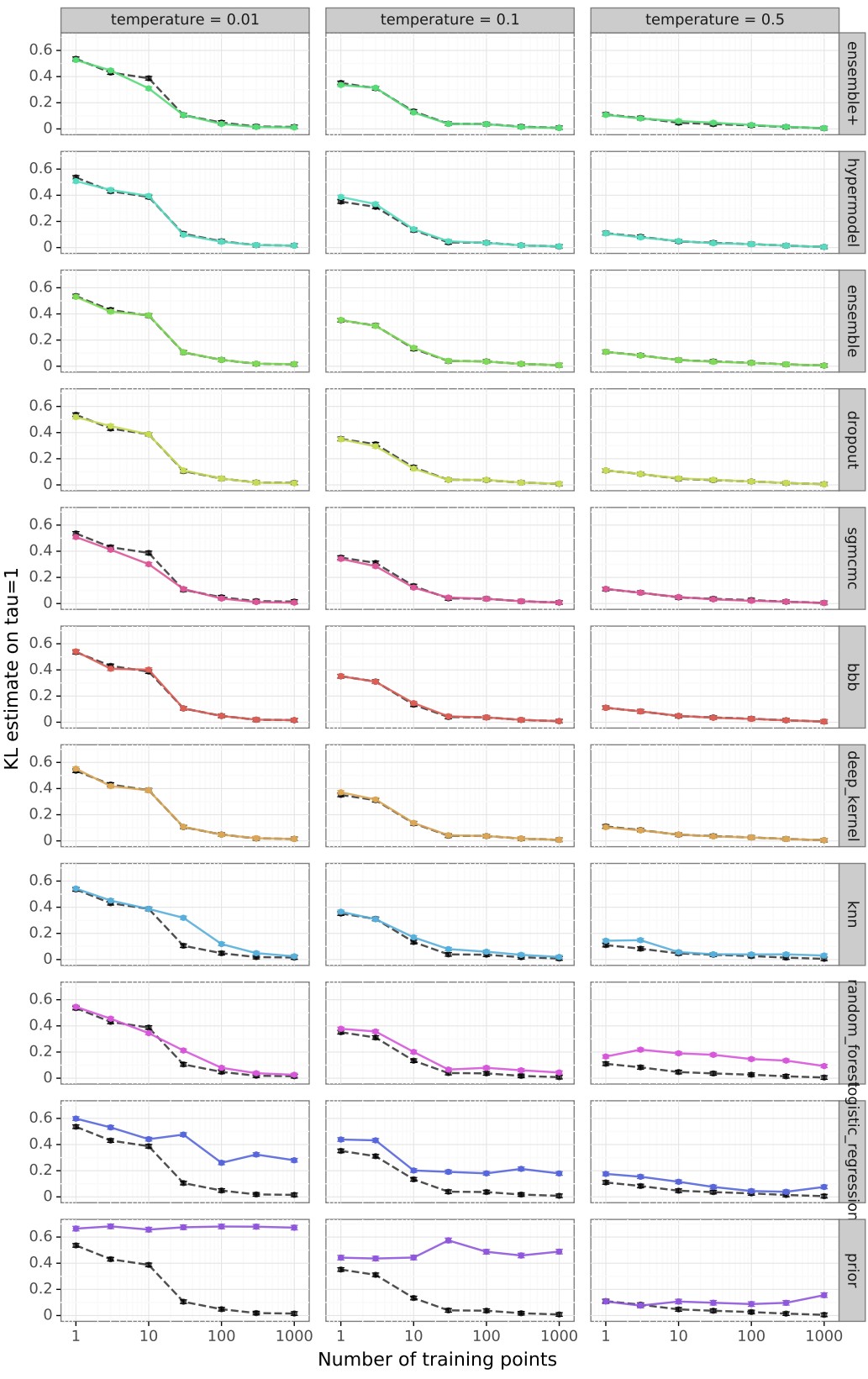

Figure 13: Performance of benchmark agents on the Testbed evaluated on $\tau = 1$, compared against the MLP baseline.

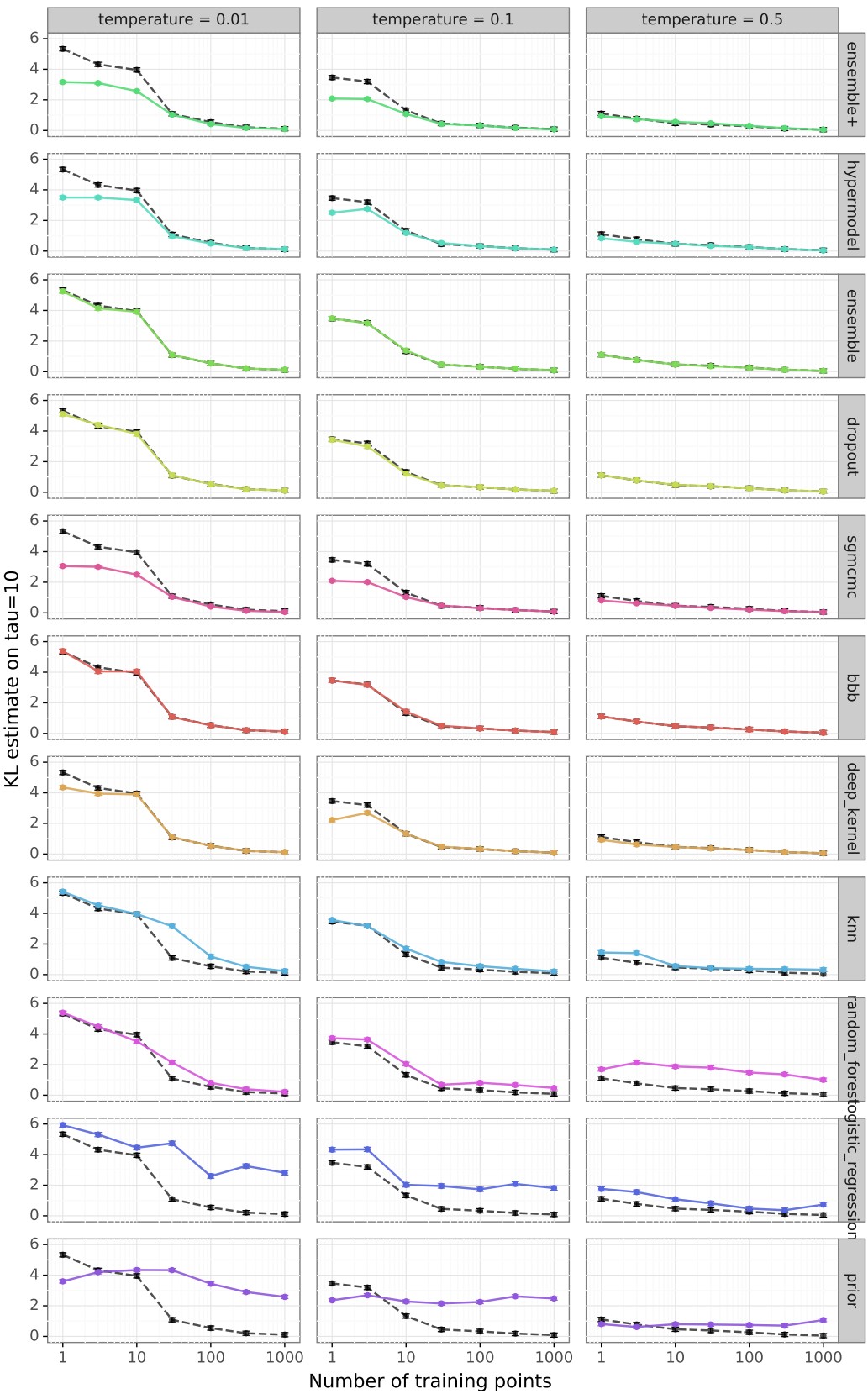

Figure 14: Performance of benchmark agents on the Testbed evaluated on $\tau = 10$, compared against the MLP baseline.

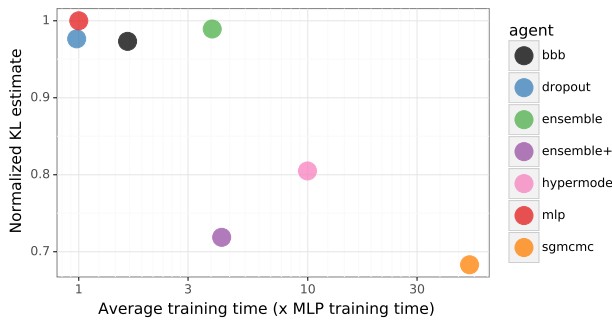

Figure 15: Normalized $\mathbf{d}^{10}_{\mathrm{KL}}$ vs training time of different agents

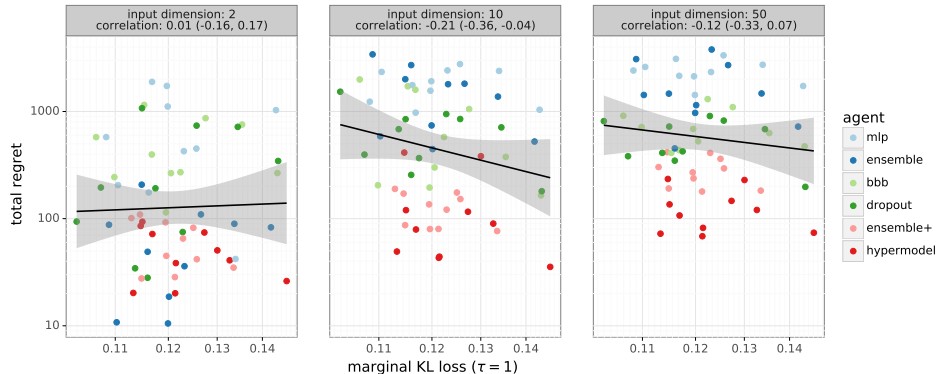

Figure 16: Testbed marginal performance $\mathbf{d}^1_{\mathrm{KL}}$ is not significantly positively correlated with sequential decision performance. This result is robust across input dimensions 2, 10, and 50.

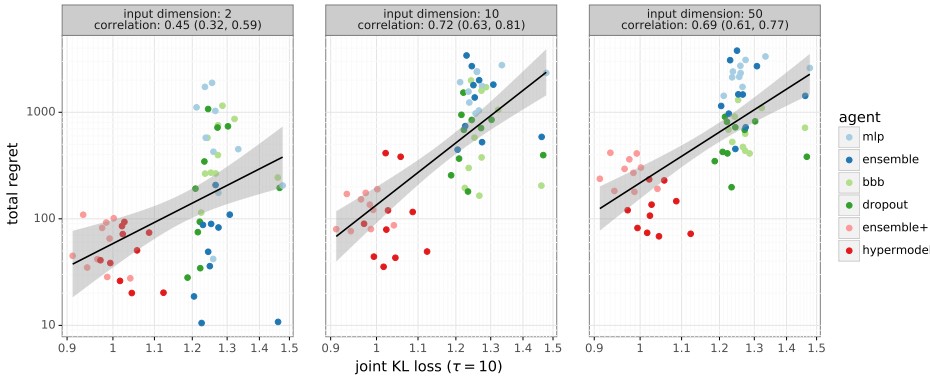

Figure 17: Testbed joint performance $\mathbf{d}^{10}_{\mathrm{KL}}$ is significantly positively correlated with sequential decision performance. This result is robust across input dimensions 2, 10, and 50.