# OpenReview forum: "The Neural Testbed: Evaluating Joint Predictions"
_NeurIPS.cc/2022/Conference — NeurIPS 2022 Accept_

### Official Review · Reviewer_4Cxf · 2022-07-10

**Rating:** 7
**Confidence:** 4
**Soundness:** 3 good
**Presentation:** 3 good
**Contribution:** 3 good

**Summary:**

This paper proposes a benchmark task to evaluate Bayesian deep learning agents on the calibration of their joint predictions rather than just marginal predictions. This benchmark generates generative processes via randomly initialized neural networks and evaluates agents based on their KL-divergence from the true generative distribution over joint predictions. A major contribution of this work is to show that in marginal predictions, none of the Bayesian methods perform significantly better than a standard MLP. However, in joint predictions, several Bayesian deep learning methods underperform. The paper also shows that this performance is correlated with performance on a neural bandit task using Thompson sampling. Finally, the paper shows that the results are fairly robust to the choice of generative model.

**Questions:**

1. How does this work compare to (Wang et al. 2021)?
2. Why do the different agents perform differently in the benchmark tasks? Is there some takeaway that the benchmark can give us that demonstrates its scientific value?

**Limitations:**

The paper talks briefly about limitations, such as the fact that the paper is meant as a sanity check rather than a grand challenge.

**Strengths And Weaknesses:**

1. Originality:  Many prior works in Bayesian deep learning include benchmarks against other methods, however they do not measure the performance of *joint* predictions. The paper mentions that a prior work (Wang et al. 2021) has a similar motivation of moving beyond measuring the quality of marginal uncertainty estimation, but that their paper shows the potential of measuring joint likelihoods. I would like to see a bit more of a comparison between the methods in that paper and in this one, and how the results compare.
2. Quality: The paper is motivated well and the experimental framework proposed makes sense. I would like to see some more intuition of *why* certain agents perform better than others in this benchmark in the main-text.
3. Clarity: The paper is very well written and motivated. The plots are easy to understand.
4. Significance: This paper gives an important and accessible benchmark for evaluating the quality of Bayesian agents.

---

> ### Author Response · Authors · 2022-07-28
> **Author response**
>
> We thank you for your efforts in the review process, and point you to our "shared response" that I think focuses on the main weaknesses you have raised in your review (and concerns shared by others). For what it's worth, we agree that these are interesting questions for this line of research, but don't believe that these constitute weaknesses in terms of this conference submission... and we hope that you would consider upgrading your rating to a stronger "accept".
>
> We also hope that you can help us in discussions with the Reviewer Ayr1, who has such negative views on the paper.
>
> In terms of your specific questions, I think the main one we should expand on is the connection to Wang et al 2021:
> http://proceedings.mlr.press/v130/wang21g/wang21g.pdf
>
> In their Section 3.3.1they consider evaluating joint log-likelihoods (as we do in this paper) but decide it is unsatisfying for two reasons:
>
> 1. Impact of predictive marginals: We’ve found the joint log-likelihood scores to be determined almost entirely  by the marginal log-likelihood scores, with only a small dependence on the PPCs. Hence, in practice, they provide little new information beyond marginal log-likelihoods.
>
> 2. Uncorrelated random batches. The points in a random batch are almost uncorrelated because they usually scatter far away from each other.
>
> Our results show that evaluating joint log likelihood directly can provide clear distinctions in agent performance.
> There are several potential explanations for this discrepancy:
>
> - They look at a different environment (regression not classification, different kernels etc)
> - They look at different agents (if we'd only included bbb, ensemble, dropout we also would not have seen a significant difference)

---

> > ### Comment · Reviewer_4Cxf · 2022-08-07
> > **Thank you**
> >
> > Thank you for the response. After reading the other reviews and the author responses, I am convinced that the authors' choice to not include significant analysis about why different agents perform differently is the correct one. I also thank the authors for helping me better my understanding of the difference between this paper and Wang et al 2021. I encourage the authors to incorporate some of these points in the related work section. I will continue to recommend acceptance for this paper and increase my confidence score.

---

### Official Review · Reviewer_Yeyc · 2022-07-11

**Rating:** 6
**Confidence:** 3
**Soundness:** 3 good
**Presentation:** 3 good
**Contribution:** 3 good

**Summary:**

The paper proposes an open source benchmark for evaluating uncertainty quantification methods. The testbed enables evaluations using marginal and joint predictions by controlling the data generation process using neural network based simulated datasets and proposes to use KL divergence as the evaluation metric. The paper also uses the testbed to provide insights on the performance of Bayesian deep learning approaches. Their analyses show that popular techniques in this category, including Monte Carlo dropout, ensembling and Bayes by Backprop do not outperform point estimates in marginal and joint predictions. They show that the performance of joint estimates is correlated with performance in neural bandits through regret analyses.


**Questions:**

- I recommend adding a discussion on the limitations of the neural testbed. From the manuscript, it is hard to understand whether the assumptions done for analyses are a limitation of the neural testbed.
- Along the same lines, even though the KL divergence metric is proposed as the core metric, it would be great to include a discussion on other commonly used metrics (like calibration and likelihood based metrics) and how neural testbed can support these.
- It would be good to explain how the users can define the environment epsilon distribution since it constitutes the main backbone of the data generation.
- The results reported in Section 4.1: It is interesting to see that probabilistic approaches do not outperform MLP in terms of calibration error, which is counter intuitive given the literature. The fact that the generative model and the model architecture match provides an explanation but it would be great to see some results (at least in an appendix) where this is not the case.
- It might be good to extend the robustness analysis to include number of layers, which can help provide a more complete picture.
- The insights on SGMCMC has significant parallels with Izmailov et al. (2021)’s findings and it might be good to include these in the discussion.

Minor comments:
- There is a typo in Algorithm 1. Add a space between “environment” and “likelihood”.
- Line 241. I recommend adding a discussion on the gist of how prior functions enable the benefit of ensemble+ to ensemble in the manuscript.
- Line 287. Missing a verb between “we will also” and “these agents”

Izmailov, P., Vikram, S., Hoffman, M.D. and Wilson, A.G.G., 2021, July. What are Bayesian neural network posteriors really like?. In International conference on machine learning (pp. 4629-4640). PMLR.


**Limitations:**

The main limitation of the paper is the toy nature of the datasets and models considered in this study. I would recommend adding a discussion on how neural testbed scales to more realistic settings. Along similar lines, the manuscript can benefit from a discussion on the general limitations of neural testbed.


**Strengths And Weaknesses:**

Originality: The methods are not new but the paper introduces a new open source framework to provide a unified approach to evaluate the existing methods. Furthermore, the extensions of such evaluations to joint predictions is quite novel to my knowledge.

Quality: The paper provides a framework for evaluation and provides extensive empirical analyses with this framework. The proposed framework is technically sound.

Clarity: The paper is well written and organized.

Significance: This is an important open source contribution that can help standardize the uncertainty quantification approaches, which is a bit scattered at this point in my opinion. Therefore, I believe this is an important contribution to the field. However, the experiments are conducted in rather small data environments with toy MLP models, which is limiting its evaluation on how it can help advance the state of the art.

---

> ### Author Response · Authors · 2022-07-28
> **Author response**
>
> We thank you for your efforts in the review process, and point you to our "shared response" that I think focuses on the main weaknesses you have raised in your review (and concerns shared by others). For what it's worth, we agree that these are interesting questions for this line of research, but don't believe that these constitute weaknesses in terms of this conference submission... and we hope that you would consider upgrading your rating to a stronger "accept".
>
> In response to your more specific questions:
>
> > I recommend adding a discussion on the limitations of the neural testbed. From the manuscript, it is hard to understand whether the assumptions done for analyses are a limitation of the neural testbed.
>
> This seems like a good idea, particularly in response to the questions shared across reviewers. In terms of "assumptions" though it would be helpful to know what specific assumptions you are asking.
>
> > Along the same lines, even though the KL divergence metric is proposed as the core metric, it would be great to include a discussion on other commonly used metrics (like calibration and likelihood based metrics) and how neural testbed can support these.
>
> Yes, we should clarfiy this, particularly where KL-divergence is actually equal to negative log likelihood plus an irreducible constant, so these are essentially the same metric (see https://arxiv.org/abs/2107.09224).
>
> For ECE, we can and do calculate this, but the differences in agent performance are not statistically significant.
> This testbed is also really designed to focus on *joint* statistics, and we are not aware of a joint-style analog to ECE.
>
> > It would be good to explain how the users can define the environment epsilon distribution since it constitutes the main backbone of the data generation.
>
> This is of course possible using our code, but actually not a major focus on our research.
> We want to encourage people to use a single benchmark model (2-layer 50 hidden unit ReLU MLP) so that results are more easily comparable across papers.
> Our results in Section 6 show that, in some sense, the results are robust to minor variations in this choice.
>
> > The results reported in Section 4.1:..
>
> See main response (2)
>
> > It might be good to extend the robustness analysis to include number of layers, which can help provide a more complete picture.
>
> We can certainly add this to an appendix.
> Including it in the main paper may end up somewhat overwhelming.
>
> > The insights on SGMCMC has significant parallels with Izmailov et al. (2021)’s findings and it might be good to include these in the discussion.
>
> Yes, we can expand on this discussion... certainly compatible.

---

> > ### Comment · Reviewer_Yeyc · 2022-08-09
> > **Response on assumptions/limitations**
> >
> > I thank the authors for their response for my feedback. Regarding my comment on limitations/assumptions: The main assumption I was referring to was around the environment epsilon distribution and its scalability aspect. This is also raised by other reviewers as the analyses are performed with simple models. Having said that, I am happy to hear that there are already work building on this work.

---

### Official Review · Reviewer_Ayr1 · 2022-07-14

**Rating:** 2
**Confidence:** 3
**Soundness:** 1 poor
**Presentation:** 1 poor
**Contribution:** 2 fair

**Summary:**

This paper proposed a new benchmark to evaluate joint predictions (with focus on Bayesian deep learning) by generating random classification problems using a generative process. While previous works largely focused on marginal prediction, this work mainly focused on joint predictions considering its importance in the decision making process. In the paper, 7 different agents (e.g. mlp, ensemble, dropout, etc.) are included. Also, the authors released the code for this paper.

**Questions:**

The paper keeps switching between deep learning and bayesian deep learning in the text (e.g. 'The Neural Testbed as a simple and clear benchmark for evaluating the quality of joint predictions in deep learning systems. ', 'we compare benchmark approaches to Bayesian deep learning', '..inform the field’s wider efforts in deep learning..', '...less transferable to general deep learning research'). While I can kinda guess why this happened, I am not sure what is the main focus of this paper? Will it be a general framework that can be used in any problem settings regardless of its solutions or it can only be suitable to evaluate bayesian approaches? It seems the latter is the case in this paper.

Where is the relevant text for Figure 3? it is neither used in the text nor even has a detailed caption.

**Limitations:**

The paper briefly discusses the limitations of their method and future direction in the conclusion. Societal impact is not relevant in this paper.

**Strengths And Weaknesses:**

The direction of this paper is very interesting and as the authors rightfully said joint predictions as an essential piece in decision making processes. The paper idea is well-motivated and figure 1 clearly shows why this question should be studied. That being said, apart from the first few sections, (unfortunately) the paper lacks a consistent story and doesn't provide details in the main text about the proposed approach, setting, how it works, etc, For instance, one needs to keep checking the appendix to understand main ideas in the paper as lots of important details are relegated to the appendix. In addition, experiments or evaluation scenarios are not clear at all. Take Figure 2 as an example, it is not clear what is the scenario, what is the task?, what is the experiment settings?, why joint prediction is relevant in this example? The same thing is true about experiments in sections 4.2, 4.1, etc. Importantly, their proposed method is not explained well and not clear e.g. how it generates random classification tasks? how data will be generated? etc. The fact that this is a benchmark paper and lacks details, is very concerning. Benchmark paper should be very generous on details and be super clear about experiment setups and procedures, unfortunately neither of them is the case in this paper. The paper needs significant revision and rewriting.

Moreover, it seems this paper is more of a conceptional benchmark rather than a real world one as all experiments/settings are rather synthetic or toy examples. Take robustness study as an example. It only considers a simple MLP with few hidden units, not sure how this result can help with different settings/architectures and what would be the conclusion here?


One of the main and natural applications of this work is to be used in sequential decision making tasks. Although the paper studies a simple and toy bandit case, realistic scenarios could have been utilized in this paper. For example, the paper could have studied the issue of safety in sequential decision making tasks, etc.

In summary, the proposed method is not clear and the paper doesn't provide a well defined picture about their method, setting, and the experiments. It is not ready in its current format and requires significant revision.

---

> ### Author Response · Authors · 2022-07-28
> **Author response**
>
> We thank you for your efforts in the review process, and point you to our "shared response" that hopefully helps to address the main concerns in this paper.
> We are disappointed that you viewed the paper so negatively compared to the other reviews.
> We hope that, through this rebuttal process, you might view the paper in a new light and increase your rating to something more positive.
>
> In response to your concerns on clarity:
> - Reviewer CzHp: This paper is well-written and easy to follow. All implementation details can be found in the provided codebase.
> - Reviewer Yeyc: The paper is well written and organized.
> - Reviewer 4Cxf: The paper is very well written and motivated. The plots are easy to understand.
>
> Of course, we want to make sure that we address specific problems you have raised, but in this case I do not believe your complaints are shared by other readers.
> In response to some of your specific questions:
>
> > how it generates random classification tasks?
>
> Section 3.1 provides an outline of the generative model.
> Unfortunately, in the interests of space we had to relegate some details to Appendix B, but this is done in the interest of clarity and the demands of a conference page limit.
>
> Of course, we also put significant effort into our opensource code release, which provides the full code necessary to reproduce all details.
> If there are specific pieces you believe we should highlight more clearly in the main paper, then of course we will work to include these.
>
>
> > Confusion between deep learning and "Bayesian deep learning"
>
> As you point out, at several places in the paper we discuss both deep learning and Bayesian deep learning.
> My belief is that there is nothing wrong or inconsistent with our use of these terms.
>
> The Neural Testbed is a synthetic deep learning prediction problem.
> We use a neural network generative model, and compare the performance of agents that make predictions in this setting.
> We also benchmark several state of the art approaches to Bayesian deep learning, which is one family of algorithms designed to make good predictions in this setting.
>
> The Neural Testbed can evaluate any learning agent that makes predictions, whether it is Bayesian or not.
> As such it elegantly sidesteps the contentious problems of whether method "XYZ is really Bayesian?".
> However, some of our key results provide insight to the efficacy of popular Bayesian deep learning approaches.

---

> > ### Comment · Reviewer_Ayr1 · 2022-08-08
> > **Re: Author response**
> >
> >
> > While I thank the authors for their responses, I stay with my previous assessment that this paper is not ready for Neurips and needs major revision.
> >
> > I strongly disagree with your response that says "This is a somewhat intentional choice in a short paper, designed to sidestep potential disagreements on contentious issues". Your paper and results should speak for themselves. We write/read (scientific) papers to argue/explain/analyze new/existing findings. We can't use "potential disagreements" as a way to refrain from explaining our findings. Also, this paper is a benchmark paper and I'd assume a benchmark paper like this should be as comprehensive as possible in terms of results and discussions.
> >
> > In addition, "toy nature of the Neural Testbed" in my view is a deal breaking issue with this paper. As I mentioned in my review, the results from toy experiments don't necessarily tell us anything about more realistic scenarios.
> >
> > Finally, the first two pages of your paper are fantastic and it is a great example of how one should motivate an idea. However, unfortunately, that is not true about the rest of the paper as lots of details are missing and things are unclear (please see my review). This is a benchmark paper, thus the proposed benchmark should have been described in meticulous detail. In my view, it would be hard for someone to read your paper and reproduce your experiments, even the toy ones.

---

### Official Review · Reviewer_CzHp · 2022-07-18

**Rating:** 6
**Confidence:** 3
**Soundness:** 2 fair
**Presentation:** 3 good
**Contribution:** 3 good

**Summary:**

This paper introduces an open-source benchmark called "Neural Testbed" to evaluate the quality of neural networks' predictive distribution. Instead of using a finite collection of datasets, the Neural Testbed adopts a synthetic data generation process to avoid overfitting. Besides, this paper advocates evaluating the quality of prediction on joint prediction rather than merely on marginal distribution. The authors find that previous Bayesian inference methods that give an excellent marginal prediction may fail to provide an accurate joint prediction, and the quality of the joint prediction is closely related to the performance on sequential decision tasks.

**Questions:**

4.1 Performance in marginal prediction

- Why do you choose aggregated KL as the objective? Does it imply there is a trade-off between joint prediction and marginal distribution? Does it mean a model that is good at $\mathbf{d}_{\mathrm{KL}}^{10}$ prediction may not perform well on marginal distribution or joint distribution with 100 examples?
- Have you verified the two hypotheses for the discrepancy? What results do you get if you tune for ECE? What happens if the architecture does not match?
- Why does the ensemble model achieve worse accuracy than the simple MLP model?
- Why do agents with better $\mathbf{d}_{\mathrm{KL}}^{10}$ have worse ECE performance?

4.2 Performance beyond marginals
- Why do some agents have better joint prediction ability, such as hypermodel, ensemble+, sgmcmc? And why do ensemble, dropout, bbb fail to do so? Figure 12 shows that diversity may be a key factor. Does this hold for the other methods? What are other hypotheses?

5.2 Sequential Decision
- It is interesting to see that the quality of joint distribution and the performance of a sequence decision task is highly correlated. But why does the sequential decision problem rely on joint prediction quality? It seems that Thompson sampling does not explicitly exploit the joint prediction. What's the explanation for this observation?

Misc
- Line 287: "we will also these agents over alternative environments," missing a verb.
- In tutorial.ipynb, it would be easier for the reader if some of the key concepts like "ENN" were explained. This abbreviation does not appear anywhere in the main text or appendix.

**Limitations:**

This paper discusses its limitation and points out some future research directions.

**Strengths And Weaknesses:**

Strength
- This paper provides a simple benchmark to evaluate the quality of distribution, which can benefit the community.
- This paper points out the importance of the quality of joint distribution. It observes a positive correlation between the joint distribution quality and the performance of sequential decision-making problems.
- This paper is well-written and easy to follow. All implementation details can be found in the provided codebase.

Weakness
- The synthetic data generation processes and model architecture are simple. It's unclear whether the results still hold on large-scale and more challenging datasets or more complex deep models.
- Lack of explanations for some key observations.

---

> ### Author Response · Authors · 2022-07-28
> **Author response**
>
> We thank you for your efforts in the review process, and point you to our "shared response" that I think focuses on the main weaknesses you have raised in your review (and concerns shared by others).
> For what it's worth, we agree that these are interesting questions for this line of research, but don't believe that these constitute weaknesses in terms of this conference submission... and we hope that you would consider upgrading your rating to a stronger "accept".
>
> To address your specific questions:
>
> > Why do you choose aggregated KL as the objective? ...
>
> This choice is somewhat arbitrary, and actually we could have selected on D_KL^10 alone to achieve similar results.
> Certainly for the agents that perform *well* in joint prediction, that also leads to performing well in marginal predictions.
> We can include alternative rankings in an appendix, but these choices make little difference.
>
> > What results do you get if you tune for ECE?
>
> None of the agents perform statistically siginficantly different in ECE according to our testbed. Our confidence bars are pretty wide on this metrics.
>
> > What happens if the architecture does not match?
>
> Section 6 focuses on this question, and shows that the results are quite robust across environment models.
>
> > Why does the ensemble model achieve worse accuracy than the simple MLP model?
>
> These results are not statistically signficant, within the margin of error.
>
> > Why do agents with better have worse ECE performance?
>
> These results are not statistically signficant, within the margin of error.
>
> > 4.2  Performance beyond marginals
>
> See our point (2) in the shared rebuttal.
>
> > 5.2 Sequential Decision
>
> This is an interesting topic for research, there are some arXiv preprints that consider this question: https://arxiv.org/abs/2107.09224.
>
> The core observation is perhaps that the future *cumulative* rewards as in some sense a joint prediction.

---

### Author Response · Authors · 2022-07-28
**Author response**

## Summary

We would like to thank all of the reviewers for their time and effort in the review process. Overall, we are pleased to see that most reviewers recognise the key contributions of this work, and recommend acceptance at NeurIPS.

In this statement, we will address the two main critiques/questions that have been raised across reviews:
1. The toy nature of the Neural Testbed generative model.
2. The lack of definitive explanation why certain agents outperform others.

We view these as somewhat intentional choices for this short workshop paper. We believe that comprehensively answering these questions, and extending this line of research could be a fruitful area of future investigation. However, we do intend to incorporate the feedback where possible to help clarify the limitations and focus of this paper.

Beyond this two “big picture” points, we will also take care to address the specific questions of each reviewer in turn. For the most part, these should be relatively straightforward to incorporate, and we thank you once again for your contributions.

We hope that, after addressing these main concerns, the reviewers will be happy to increase their scores and recommend acceptance at the conference. In the special case of the reviewer who gave us a “strong reject”, we hope that this rebuttal period, and the perspectives of other reviewers may lead you to reconsider your position.

---

> ### Author Response · Authors · 2022-07-28
> **1. The toy nature of the Neural Testbed generative model**
>
>
> ## 1. The toy nature of the Neural Testbed generative model.
>
> Several reviewers have pointed out that the Neural Testbed focuses on a simple/toy generative model: random 2-layered MLPs.
> We actually agree with this assessment, but do not view that as a weakness of our submission.
> As reviewer 4Cxf discusses, this is “meant as a sanity check rather than a grand challenge”.
> We explicitly call out this motivation in several places of the paper:
> Lines 39-41: “sanity check … in a simple setting”
> Line 66: “a simple benchmark for the field”
> Line 148: “a simple, clear and accessible testbed can provide significant value to community”
> Line 230: “even in this toy setting”
> Line 280: “natural concern… less transferable to general deep learning”
> Line 314: “are there analogous results to Figure 2 on large-scale datasets?”
>
> So yes, we agree that the generative model is extremely simplified. It is not clear whether the same observations will extend to modern large deep learning settings. However, it is interesting that even in this simple setting, we are able to obtain such drastically different performance from several benchmark Bayesian deep learning models, which should be well-suited to this domain.
>
> This is a novel result to the community, and should be interesting to many researchers in the field. In fact, we view that these results occur in such a simple and foundational setting as a major positive for this work. Our results in Section 6 show that the high-level takeaways in both joint prediction and decision performance are very robust to the precise choices of generative model.
>
> There are already some arXiv preprints that build upon the work in the Neural Testbed, and seek to extend this sort of analysis to larger-scale challenge datasets. As cited in our submission line 305, these papers claim that the general patterns from this simple testbed can extend to real datasets in some settings.
>
> That our simple testbed can unearth such drastic differences in benchmark agent performance brings us to the second main question from the reviewers:

---

> ### Author Response · Authors · 2022-07-28
> **2. The lack of explanation why certain agents outperform others**
>
> In our paper, we present the clear and reproducible results that several of the most popular, and widely cited, approaches to Bayesian deep learning do not significantly outperform a well-tuned MLP in this simple setting. This is a really interesting and potentially contentious result, that should be of significant interest to the community.
>
> However, as the reviewers have pointed out, we do not provide definitive answers for why these methods (dropout, ensemble, bbb) are not performing well in joint prediction compared to the “gold standard” of SGMCMC. This is a somewhat intentional choice in a short paper, designed to sidestep potential disagreements on contentious issues where we don’t necessarily have all the answers (or space) to definitively answer these questions. We want to raise this as an issue, and leave that for future work.
>
> For example, even though “dropout as a Bayesian approximation” has over 5k citations, there are several other rarely-cited papers that claim it is a very poor approximation:
> - Osband 2016: http://bayesiandeeplearning.org/2016/papers/BDL_4.pdf
> - Hron 2017: http://bayesiandeeplearning.org/2017/papers/45.pdf
>
> Similar disagreements erupt regularly on twitter and in conferences about whether method “XYZ is really Bayesian?”, and we wanted to avoid this type of contentious debate during our review process. Our paper sticks to simple, well-supported facts around performance on the Neural Testbed, and we hope that future work can extend on this intuition.
>
> Since, as far as we know, none of the Bayesian deep learning papers have actually claimed to have good joint predictive performance (since this is not a metric that has attained much focus in the community thus far). As such, our results that show that these agents don’t perform well, don’t even contradict the previous claims that the agents perform well at marginal prediction in Bayesian deep learning.
>
> In terms of building an intuition for our readers, we do include an Appendix D.1, which visualizes the predictions of ensemble vs ensemble+ agents. This seems to provide some insight that tracks well with the claims of the “randomized prior functions” paper… and we will call this out more explicitly in the main paper.

---

### Meta-Review · Area_Chair_wYKX · 2022-08-31

**Recommendation:** Accept
**Confidence:** Certain

**Metareview:**

Getting a reasonable estimation of joint predictions is crucial for many uncertainty estimation tasks. The paper proposed a set of benchmarks for predicting joint probabilities of the outputs over a few input examples. The proposed synthetic tasks are easy to deploy and test on most Bayesian methods, including Bayesian neural networks.

Uncertainty estimation is one of the fundamental challenges for modern machine learning algorithms. Many downstream application areas in reinforcement learning, active learning, and safety require a model to assess its prediction confidence. Yet, unlike the standard classification tasks, there is a lack of benchmark datasets to evaluate the performance of uncertainty methods. The strength of this paper is:
1) Develop a suite of benchmarks, although synthetic and toyish, to allow a quantitative study of the joint prediction of the existing machine learning methods. The proposed benchmark allows researchers to study uncertainty estimation without invoking any downstream application in RL or active learning.
2) The work bridge the gap between the benchmarks on marginal predictions, such as Riquelme et al. "Deep Bayesian bandits showdown" and heavy machinery of exploration tasks in RL.
The weakness of the current submission is a lack of clarity in the current writing, as pointed out by one of the reviewers. Many experimental hyperparameters are omitted from the main text, which would help the readers understand the benchmark details and design choices. Also, there is a glaring limitation of the benchmarks' simplicity and whether the generative model choice could generalize to high-dimensional problems.

Given the scarcity of other benchmarks in the uncertainty estimation tasks, the strength outweighs the weakness of this paper.

**Award:**

No

---

### Decision · Program_Chairs · 2022-09-14

Accept